# Comparative Transcriptome Analysis of Two *Kalanchoë* Species during Plantlet Formation

**DOI:** 10.3390/plants11131643

**Published:** 2022-06-22

**Authors:** Francisco Jácome-Blásquez, Joo Phin Ooi, Leo Zeef, Minsung Kim

**Affiliations:** 1School of Biological Sciences, Faculty of Biology, Medicine and Health, University of Manchester, Manchester M13 9PT, UK; francisco.jacomeblasquez@outlook.com (F.J.-B.); joophin.ooi@outlook.com (J.P.O.); 2Bioinformatics Core Facility, Faculty of Biology, Medicine and Health, University of Manchester, Manchester M13 9PT, UK; leo.zeef@manchester.ac.uk

**Keywords:** asexual reproduction, plantlet formation, organogenesis, embryogenesis, gene ontology

## Abstract

Few species in the *Kalanchoë* genus form plantlets on their leaf margins as an asexual reproduction strategy. The limited molecular studies on plantlet formation show that an organogenesis ortholog, *SHOOTMERISTEMLESS* (*STM*) and embryogenesis genes, such as *LEAFY COTYLEDON1* (*LEC1*) and *FUSCA3* are recruited during plantlet formation. To understand the mechanisms of two *Kalanchoë* plantlet-forming species with different modes of plantlet formation, RNA-sequencing analysis was performed. Differentially expressed genes between the developmental stages were clustered in *K. daigremontiana* (Raym.-Hamet and H. Perrier) and *K. pinnata* (Lam. Pers.), respectively. Of these gene clusters, GO terms that may be involved in plantlet formation of both species, such as signaling, response to wounding, reproduction, regulation of hormone level, and response to karrikin were overrepresented. Compared with the common GO terms, there were more unique GO terms overrepresented during the plantlet formation of each species. A more in-depth investigation is required to understand how these pathways are participating in plantlet formation. Nonetheless, this transcriptome analysis is presented as a reliable basis for future studies on plantlet formation and development in two *Kalanchoë* plantlet-forming species.

## 1. Introduction

Flowering plants (angiosperms) reproduce sexually or asexually in numerous and complex ways [1,2,3]. Selective preference for sexual or asexual reproduction and the specific strategy used depends on the species and external environmental conditions [3]. Asexual reproduction is particularly advantageous when a species is already in a habitat with favorable conditions as it allows the progeny to integrate earlier into existing populations [1,4]. Plants have developed several different forms of asexual reproduction, including apomixis, stolon, corms, rhizomes, and adventitious buds [4]. Other strategies of vegetative reproduction consist of the formation of new plants connected with the mother plant by tubers, rhizomes or stolons [5]. Although asexual reproduction strategies are common among perennial plants, molecular and genetic mechanisms controlling asexual reproduction are still elusive.

*Kalanchoë* developed a unique asexual reproduction strategy [6]. Several *Kalanchoë* species acquired the ability to reproduce asexually by forming new plants (plantlets) on the margins of leaves. Phylogenetic analyses of *Kalanchoë* revealed an evolutionary trajectory of the plantlet formation strategies. While *Kalanchoë* species in the basal group, such as *Kalanchoë tomentosa* and *K. marmorata*, are unable to form plantlets in leaf margins [7,8], species in derived clades, such as *K. daigremontiana* and *K. tubiflora,* constitutively form plantlets on the margins under long-day conditions [7,9]. Intriguingly, an evolutionary transition is observed in the clades between basal and derived clades. *Kalanchoë* species in these transitory clades (e.g., *K. fedschenkoi*, *K. prolifera*, *K. pinnata*, *K. streptantha,* and *K. gastonis-bonnieri*), form plantlets after leaves are severely damaged or detached from the mother plant, and thus is referred to as inducible plantlet formation [6,7]. Interestingly, in *Kalanchoë*, asexual reproduction strategies *via* plantlet formation might have developed as a trade-off with regular sexual reproduction. While inducible- and non-plantlet-forming species produce viable seeds, constitutive-plantlet-forming species generate non-viable seeds and have lost the ability to reproduce sexually [7,10]. While plantlet formation in constitutive and inducible species appears to be superficially similar, the developmental mechanism(s) in these *Kalanchoë* groups are suggested to be different The *K. pinnata* inducible-plantlet-forming species develops epiphyllous buds during leaf formation, which remain dormant until leaves are excised from the plant, presumably due to the disrupted hormone supply which triggers shoot initiation [11,12]. In some cases, bud dormancy does not affect root initiation as adventitious-like roots emerge when leaves are still attached to the plant [10]. On the other hand, plantlet formation in the *K. daigremontiana* constitutive-plantlet-forming species resembles zygotic embryogenesis, but skips dormancy and the seedling stage [13,14]. In contrast with inducible-plantlet-forming species, constitutive-plantlet-forming species form pedestal structures in the notches, from which plantlet primordia emerge. Once the plantlets are fully formed, excision sites are established at the base of the plantlet through a programmed cell death pathway, allowing them to fall on the ground and start growing independently [7,14].

To date, little is known regarding the genetic mechanism(s) modulating plantlet formation in *Kalanchoë* leaves. Ectopic expression of class 1 *KNOTTED-LIKE HOMEOBOX1* (*KNOX1*) gene, *SHOOTMERISTEMLESS* (*STM*) in leaf notches was proven to be required for the formation of plantlets. *STM*-downregulated *K. daigremontiana* transgenic plants were unable to develop plantlets [7]. *STM* is normally expressed in the shoot apical meristem (SAM), where a constant population is required to maintain pluripotent cells [15]. In *Arabidopsis*, *STM* is required to develop and maintain a functional SAM and its ectopic expression induces de novo meristem formation [16]. The *stm*-silenced *Arabidopsis* mutants failed to organize a SAM during embryogenesis [17]. Consistent with the fact that *STM* expression on the leaves of *Kalanchoë* plantlet-forming species is required for plantlet formation, *STM* was not expressed in leaves of non-plantlet-forming species [7]. This suggests that plantlet formation is facilitated by meristematic pathways. In addition, two embryogenesis genes, *LEAFY COTYLEDON1* (*LEC1*) and *FUSCA3* (*FUS3*), which were found to be expressed in the leaf margin and plantlet primordia of *K. daigremontiana* constitutive-plantlet-forming embryogenesis genes, were expressed [7]. The downregulation of *LEC1* did not affect plantlet formation in transgenic *K. daigremontiana* since LEC1 protein was truncated and unfunctional. Moreover, this allows plantlet primordia to bypass dormancy, and produces unviable zygotic seeds [13]. A functional *Arabidopsis thaliana LEC1* expressed in *K. daigremontiana* provided plantlet primordia seed-like traits, going through dormancy and accumulating oils, ultimately impeding normal plantlet development in the leaves [13].

RNA-sequencing analyses allow the identification of novel genes related to specific processes or pathways, quantification of gene expression under different conditions, visualization of expression trends, and comparison of transcriptomes between different species and cultivars in model and non-model plants [18,19,20]. Comparative approaches of transcriptomic analyses have been performed in evolutionary, crop yield performance and specific trait studies. RNA-sequencing analysis in crested wheatgrass (*Agropyron cristatum* L.) provided a robust molecular basis for floral initiation and development, identifying 113 flowering-time-associated genes, 123 *MADS*-*box* genes, and 22 *CONSTANS*-*LIKE* (*COL*) candidate genes [21]. In grape (*Vitis vinifera*), RNA-sequencing analysis allowed the detection of differentially expressed transcripts related to gibberellic acid (GA) and abscisic acid (ABA) pathways during paradormancy, endodormancy, and summer budding [22]. Moreover, RNA-sequencing analysis was implemented to compare transcriptomes from wild *Agave deserti*, *A. sisalana,* and domesticated *A. tequilana*, to track phylogenetic relationships and trait evolution, along with finding numerous key candidate genes modulating fructan, fibre, and stress response-related pathways [23].

This study aims to detect biological processes and genes involved during plantlet formation on constitutive (*K. daigremontiana*) and inducible (*K. pinnata*) *Kalanchoë* species using RNA-sequencing. Based on the differences in mode of plantlet formation and morphological structure of plantlets, we selected specific plantlet stages and time points to harvest tissues for our experiment. Our study serves as a pioneering source of molecular insight into plantlet formation and the development of the plantlet primordia in *Kalanchoë* plantlet-forming species.

## 2. Results

### 2.1. Morphology of Plantlet Formation and Clustering of Samples from Selected Plantlet Formation Stages and Time Points

The process of plantlet formation in *K. daigremontiana* and *K. pinnata* is superficially analogous, as both species can produce progeny from epiphyllous buds and pedestals located on the leaf margins (Figure 1A). Prior to the formation of *K. daigremontiana,* from the leaf notch localized between leaf serrations (Figure 1A(S1)), the plantlet was morphologically visible, and a pedestal was formed (Figure 1A(S2)). As the pedestal continued to develop, a plantlet primordium grew and emerged from the pedestal (Figure 1A(S3)). As the plantlet matured and formed cotyledons, it remained positioned on the pedestal (Figure 1A(S4)). Once the plantlets were fully formed, they detached from the pedestals. In the case of *K. pinnata*, the plantlet primordium emerged from a bud-like structure (Figure 1B(S1)). As the plantlet developed and formed cotyledons, the bud-like structure became less visible (Figure 1B(S2)). Eventually, roots started to grow out of the plantlet base (Figure 1B(S3)). Beyond this stage, leaves and roots continued to develop as the plantlet matured (Figure 1B(S4)). The plantlets remained attached to the senescent leaves until the leaves decomposed.

Molecular studies on plantlet formation have been limited. Therefore, RNA-sequencing analysis was conducted to capture genes and biological processes involved in the initiation and development of plantlets in *K. daigremontiana* and *K. pinnata*. Although genomes are not available for these two species, we were able to successfully map 90% of the RNA-sequencing reads to the *K. fedtschenkoi* genome (Appendix A). Approximately 50% of the reads could be counted into exons and genes for further analysis (Appendix A). Principal components analyses (PCA) plots revealed that PC1 and PC2 captured most of the variance among the samples and showed tight clustering of replicates, indicating distinct gene expression profiles between the stages studied (Figure 1C,D). In *K. daigremontiana*, there was longitudinal separation along PC1, in accordance with the stages. PC2 showed separation between the pedestal (S1 and S2) and plantlets (S3 and S4). On the other hand, for *K. pinnata,* the PC1 suggested that major changes occurred within the first 24 h after leaf detachment (samples 0hN, 4hN, and 24N). PC2 separated leaf mid-section from leaf notch samples.

### 2.2. Heatmap and Graphical Representation of the Expression of Genes in Different Clusters during Plantlet Formation

Transcriptomic analyses revealed a total of 4594 genes in *K. daigremontiana* and 5706 in *K. pinnata*, which were significantly differentially expressed during plantlet formation between stages and time points after leaf detachment. Genes with similar expression profiles were hierarchically clustered with eight clusters for *K. daigremontiana* and 12 for *K. pinnata* (Figure 2). At first glance, *K. daigremontiana* clustering is dominated by the control condition (leaf margin), which is in contrast with the plantlets. The stages S1–S4 tend to show a linear progression in up- or downregulation. This is in agreement with the PCA for *K. daigremontiana,* which shows similar linearity in gene regulation. In the control samples (C), *K. daigremontiana* gene clusters 1, 2, and 6 showed the lowest expression levels, followed by clusters 8, 7, and 4, whereas gene clusters 3 and 5 showed the highest expression levels (Figure 2A). At stage S1, gene clusters 1 and 7 showed the highest expression level, followed by clusters 2, then 3 and 4, and 5, whilst clusters 4 and 8 had the lowest expression level. At stage S2, genes in clusters 1, 2, 6, and 7 exhibited higher expression levels than the other clusters. However, at later stages, S3 and S4, genes within each cluster exhibited greater differences in expression level, as suggested by a display of different colors along the spectrum. In contrast with *K. pinnata*, this was observed in all gene clusters across most time points (Figure 2B). In *K. pinnata*, the highest expression level was in cluster 1 at 48 hM; cluster 3 at 4 hN; cluster 11 at 0 hN, and cluster 12 at 48 hN. This was in contrast with *K. daigremontiana*, as the highest expression level occurred at the same stage C (Figure 2A). Gene clusters in *K. pinnata* with the lowest expression level were gene clusters 4 at 4 hN, clusters 6 and 8 at 48 hM, and cluster 10 at 0 hN (Figure 2B). Apart from gene cluster 12 at 48 hN, the expression level of all gene clusters at 24 hN and 48 hN was fairly similar.

Gene clusters with similar expression patterns during plantlet formation in *K. daigremontiana* and *K. pinnata* were contrasted on the same graph (Figure 3). Clusters 1, 2, and 7 in *K. daigremontiana* showed a noticeable upregulation between Con and S1, but their expression decreased gradually during S3 and S4 (Figure 3A). This expression pattern was similar to clusters 1 and 7 in *K. pinnata* (Figure 3E). *K. pinnata* clusters 3 and 6 initially behaved similarly, but there was an upregulation between 24 and 48 h after leaf detachment (Figure 3I). In *K. daigremontiana*, expression of gene clusters 4, 5, and 8 dramatically dropped between Con and S1, but were upregulated again after S1 (Figure 3B). A similar expression pattern was seen in clusters 8 and 12 in *K. pinnata* (Figure 3F,J). Moreover, clusters 4 and 9 showed a similar expression pattern during the first few time points, but the genes were downregulated again 24 h after leaf detachment (Figure 3F). A steady upregulation was observed in the expression of genes in cluster 6 of *K. daigremontiana*, in which the same observation applies to clusters 5 and 10 in *K. pinnata* (Figure 3C,G). In cluster 3 of *K. daigremontiana* and clusters 2 and 11 of *K. pinnata*, an opposite expression trend was observed, where a progressive downregulation was seen from the starting point to the last stage (Figure 3D,H).

### 2.3. Number of Significantly Differentially Expressed Genes during Plantlet Formation

Among the significantly differentially expressed genes (DEGs), shared and unique expression was detected between the different stages (Figure 4) in each species. The largest number of uniquely expressed genes was observed at the initiation of plantlet formation, between stages 1 and 2 in *K. daigremontiana* (n = 3104) (Figure 4A), and in the first 4 h after leaf detachment in *K. pinnata* (n = 302) (Figure 4B). This indicated that the initiation of plantlet formation for both species involved more DEGs than at the later stages. In *K. pinnata*, there was also a large number of DEGs within 48 h after leaf detachment in notches and mid-section of the leaves (n = 560). However, this was more likely to represent genes that are important for the development of different tissues rather than for plantlet initiation, as there was no plantlet formation in the mid-section of leaves. In *K. daigremontiana*, 3104 genes were unique to S1 vs. Con, 84 genes to S2 vs. S1, and 183 genes were shared between Con, S1, and S2. Seventy nine genes were unique to S3 vs. S2 and 79 genes to S4 vs. S3. From these, 15 genes were shared between S2, S3, and S4. Moreover, 115 genes were shared between all the stages in *K. daigremontiana*. In *K. pinnata*, 302 genes were unique to 4 h vs. 0 h, 273 genes to 24 h vs. 4 h, and 259 genes were shared between 0, 4, and 24 h. Two hundred and nine genes were unique to 24 h vs. 0 h, 156 genes to 48 h vs. 0 h, and 198 genes were shared between 0, 24, and 48 h. Five hundred and sixty genes were unique to 48 h vs. 48 h and 70 genes were shared between all the time points. To investigate shared and unique genes for plantlet formation between *K. daigremontiana* and *K. pinnata*, we compared DEGs in earlier stages of plantlet formation in these species (Figure 4C). Between the two species, 114 genes were overlapped, while 2179 genes were unique to *K. pinnata* and 2316 to *K. daigremontiana*. Among *K. daigremontiana* specific DEGs, 2013 were unique to Con vs. S1, 84 genes were unique to S2 vs. S1, and 219 genes were shared between Con, S1, and S2. Among *K. pinnata* specific DEGs, 1116 genes were unique to 4 h vs. 0 h, 588 genes were unique to 24 h vs. 4 h, and 475 genes were shared between 0, 4, and 24 h.

### 2.4. Statistical Significance of Changes in Gene Expression during Plantlet Formation

The RNA-sequencing experiment captured many transcripts during plantlet formation in both *K. daigremontiana* and *K. pinnata*. However, only certain proportions of genes exhibited statistically significant changes in expression (Figure 5). Both species showed a similar amount of up- and downregulated genes between the first two samples (Figure 5A,E). Some of these genes had a higher statistical significance in *K. daigremontiana* compared with *K. pinnata,* as seen from the presence of DEG with a −log_10_(*p*-value) of more than 60, which is higher than the most significant DEG in *K. pinnata*. The overall symmetry of the volcano plots showed slightly different tendencies in significant DEGs between the two species. Across all comparisons, there were more upregulated genes than downregulated genes in *K. daigremontiana* (Figure 5B–D), whereas in *K. pinnata*, there were similar numbers of up- and downregulated genes (Figure 5E,F,J–L). In *K. daigremontiana*, there was a larger proportion of upregulated genes when comparing gene expression between S1, S2, and S3 (Figure 5B,C) and there were few expression changes between S3 and S4 (Figure 5D). On the other hand, in *K. pinnata*, between 4 and 24 h after leaf detachment (Figure 5F), there was a similar amount of significantly up- and downregulated genes. The expression of almost all genes did not change significantly between 24 and 48 h after leaf detachment (Figure 5G). However, when comparing the control between 24 and 0 h (Figure 5J,K), up- and downregulation of genes were noticed, similar to what was observed in 0 and 48 h. Moreover, we compared DEGs for the mid-section of the leaf 48 h after leaf detachment, with roughly the same amount of up- and downregulated genes (Figure 5L). This suggests that overall, for both species, gene expression changes were more significant in the early stages or time points than in later stages.

### 2.5. Biological Processes during Kalanchoë Plantlet Formation

A list of overrepresented GO terms that overlap between gene clusters and their corresponding expression trend during plantlet formation in *K. daigremontiana* (Kd) and *K. pinnata* (Kp) is shown in Appendix A. The GO terms ‘response to stimulus’, ‘cellular process’, ‘developmental process’, and ‘multicellular organismal process’ were the most overrepresented biological processes showing the same expression trends across different clusters for both species.

### 2.6. Specific Functions of DEGs in Selected Biological Processes

Upon further analysis, we recorded the function of DEGs in the GO terms shown in Figure 6 and Appendix A. The functions described for each gene were included based on their relevance to the GO term and possible participation in plantlet formation. Therefore, the list of functions is not exclusive. Most of the genes in ‘signaling GO:0023052′ play a role in response to different types of stress. The biotic stress described includes pathogen, disease, and wounding, and the abiotic stress includes heat, salt, drought, oxidative, and endoplasmic reticulum (ER) stress. Among these genes, few were involved in regulating abscisic acid (ABA) signaling during seedling germination and growth, such as *CALMODULIN 5* (*CAM5*), *PYRABACTIN RESISTANCE-LIKE 4* (*PYL4*), *DEHYDRATION-RESPONSIVE ELEMENT-BINDING 2C* (*DREB2C*), *RING AND DOMAIN OF UNKNOWN FUNCTION 1* (*RDUF1*), *PROTEIN PHOSPHATASE 2C FAMILY PROTEIN 5* (*PP2C5*), *RING-H2 FINGER A2* (*RHA2A*), *C2 DOMAIN PROTEIN* (*C2*), and *CBL-INTERACTING SERINE/THREONINE-PROTEIN KINASE 11* (*CIPK11*). For the GO term ‘response to wounding GO:0009611’, genes belonging to the same clusters Kd2 and Kp3 were also overrepresented in the GO term ‘signaling GO:0023052’. Therefore, some of these genes are also involved in similar stress responses. *PATHOGENESIS-RELATED PROTEIN 3* (*PR3*) and *JASMONIC ACID CARBOXYL METHYLTRANSFERASE* (*JMT*) were overrepresented in both ‘signaling GO:0023052’ and ‘response to wounding GO:0009611’. Most of the genes overrepresented in ‘response to wounding GO:0009611′ are involved in jasmonic acid (JA) synthesis or dependent on JA signaling. Overrepresented genes in the GO term ‘reproduction GO:0000003’ function as the name suggests, in reproduction, which includes the formation of reproductive structures, such as flowers, siliques, pollen tubes, and seeds. The GO term ‘regulation of hormone levels GO:0010817’ contained 12 overrepresented genes, in which all of the genes except for three are associated with the plant hormone auxin.

### 2.7. Unique GO Terms in K. daigremontiana and K. pinnata Plantlet Formation

Apart from recording GO terms and genes that were overrepresented in both species, we also recorded overrepresented GO terms that were unique to one *Kalanchoë* species or the other, as shown in Appendix A. The table presents the cluster in which these terms were overrepresented and the expression trend of genes in each cluster. More unique GO terms were overrepresented in *K. pinnata* compared with *K. daigremontiana*. Genes in *K. pinnata* unique GO terms exhibited all trends of expression that is observed in our dataset, whereas in *K. daigremontiana*, the genes only showed expression trends 1, 3, 5, and 7. Moreover, a wide range of gene counts existed in these GO terms and no association was present between the gene count and overrepresented GO terms.

## 3. Discussion

This study is in contrast with the transcriptome during plantlet formation in *K. daigremontiana* and *K. pinnata*, which represent *Kalanchoë* constitutive- and inducible-plantlet-forming species, respectively. Plantlet formation in both species seems to be analogous. However, it has been proposed that constitutive-forming-species recruit an embryogenesis programme, whereas inducible-forming-species carry out organogenesis initiation after induction [7]. This was established based on molecular evidence, such as the expression of embryonic genes *LEC1* and *FUS3* on the leaf notches of *K. daigremontiana*, and the resemblance of triggering apical meristematic competence with the expression of *STM* in constitutive- and inducible-plantlet-forming species [7,13]. In the case of inducible-plantlet-forming species, plantlets originate from pre-existent primordia located on the leaf crenulations, which remain dormant possibly under a hormonal influence until external stimuli break that dormancy [24]. In the case of *K. daigremontiana*, plantlets emerge from pedestals, where somatic embryos are formed and then develop into new plants [10]. These studies suggested that plantlet formation mechanism(s) in each species may differ. However, studies were limited to only a few genes or anatomical analyses. To better identify the genetic mechanism of plantlet formation for both species, we performed an RNA-sequencing analysis that allowed us to identify biological processes and differentially expressed genes, which may play important roles in modulating the asexual reproduction of *K. daigremontiana* and *K. pinnata*. Little is known regarding the mechanism of embryonic and meristematic competency acquirements that led to a successful asexual reproductive strategy. These two species are suitable models to study somatic embryogenesis and organogenesis, as well as other developmental processes, such as dormancy and stress response.

Most of the genes represented in ‘signaling GO:0023052’ are overlapped between *K. daigremontiana* cluster Kd2 and *K. pinnata* clusters Kp1 and Kp3. In accordance with the GO database AmiGO2, this specific GO term ‘signaling GO:0023052′ includes genes involved in signal transduction within a biological system. These clusters exhibited a similar expression trend, in which there was an upregulation followed by a downregulation, suggesting that genes in these clusters might be important for the initiation or early stages of plantlet formation in both species (Appendix A, Figure 6A). In terms of the functions of these genes, most of them are involved in regulating and sensing biotic and abiotic stress (Appendix A). This observation is predictable as the rate of plantlet formation in *K. daigremontiana* is enhanced by drought stress [25]. Additionally, few plants developed a preference for asexual reproduction under certain stress conditions [3]. Moreover, *K. daigremontiana* plantlet formation is triggered only under long-day conditions [14,26]. With evidence of light-dependent selective preference for asexual or sexual reproduction [3], light or light-associated stress response might be regulating plantlet initiation of *K. daigremontiana*. Apart from that, the likely mechanism for plantlet formation in *K. daigremontiana*, somatic embryogenesis, is a consequence of stress response [27]. Existing research has shown that various stresses, such as osmotic stress, oxidative stress, heavy metal stress, temperature, and nutrients play a role in stimulating somatic embryogenesis [28,29,30,31]. Somatic embryogenesis involves the induction of totipotency or embryogenic competence of differentiated plant cells [27]. Based on the fact that embryogenesis genes *LEC1* and *FUS3* are expressed at the leaf margin of *K. daigremontiana* mother leaves and that plantlets exhibit embryo-like morphological features during early development, it was postulated that differentiated cells at the leaf notches undergo somatic embryogenesis to develop into plantlets [7,32].

Although all of the *Kalanchoë* plantlet-forming species express the *STM* gene, which is responsible for embryonic shoot meristem specification in inducible-plantlet-forming species, such as *K. pinnata*, the expression of *LEC1* is absent from the leaves [7]. This led to the speculation that plantlet formation of inducible-plantlet-forming species is primarily regulated via organogenesis, as *LEC1* is known as the master regulator of late embryogenesis. Another study on *K. laetivirens* constitutive-plantlet-forming species revealed the presence of pre-existing stem cells at the site of plantlet formation and expression of *WUSCHEL*, the regulator of stem cell homeostasis at different stages of plantlet formation [33]. Similar to somatic embryogenesis, stem cell signaling also involves phytohormones and reactive oxygen species, which contribute to oxidative stress if present at an elevated level [34]. Moreover, although a detailed mechanism of stress-induced plantlet formation has yet to be elucidated, oxidative stress was shown to affect *K. pinnata* plantlet formation via nitric oxide [35]. In the *K. tubiflora* constitutive-plantlet-forming species, the antioxidant defence of plantlets was reduced compared with mother plant leaves [36]. Moreover, the same study showed that plantlets invest more energy to prevent water loss. Osmotic stress was shown to affect plantlet formation, as the addition of sucrose prevailed cytokinin inhibition of *K. marnierianum* in vitro plantlet formation [37]. Furthermore, even though the direct impact on plantlet formation was not illustrated, increased drought tolerance and water stress that influence the survival of *Kalanchoë* plants is expected to contribute to stimulation or inhibition of plantlet formation [38,39]. Induction of *K. pinnata* plantlet through leaf ageing or detachment [11,40,41] might also be achieved through activating senescence, which occurs via integration of stress signals [42,43,44].

From the list of genes represented in ‘signaling GO:0023052’, the following genes (*CAM5*, *PYL4*, *DREB2C*, *RDUF1*, *PP2C5*, *RHA2A**, C2,* and *CIPK11)* act via regulating abscisic acid (ABA) signaling during seed germination and seedling growth. This provides evidence that ABA might be involved in the maintenance of plantlet dormancy, even though a previous experiment showed that the application of ABA did not release plantlets with functional *KdLEC1* from dormancy. The authors suggested that the window for ABA signaling to act on plantlet dormancy might be narrow [13]. However, our data suggest that ABA signaling is regulated by multiple genes. Therefore, this might explain the fact that the application of ABA is not sufficient to bypass the complexity of ABA signaling regulation to exert its effect on plantlet formation. These results indicate that the plantlet developmental stages of the tissues may be analogous to seed germination and seedling growth. Observing the association of stress response during this stage is not expected, as seeds need to be sensitive to their environment to ensure that conditions are optimal to induce germination.

The overrepresented genes in the GO term ‘response to wounding GO:0009611’ are overlapped between *K. daigremontiana* clusters Kd2 and Kd6 and *K. pinnata* cluster Kp3. Genes in these clusters were upregulated only at the earliest stages, and then genes in clusters Kd2 and Kp3 decrease their expression. However, genes in Kd6 continued to upregulate until later stages of the plantlet development (Appendix A, Figure 3). Among these genes in this GO term, two genes, *ATP-BINDING CASSETTE G11* (*ABCG11*) and *GASSHO1* (*GSO1*) provide a clue to explain the participation of these genes during early plantlet formation. *GSO1* encodes an embryonically expressed receptor kinase that is essential for embryonic cuticle formation [45,46], whereas *ABCG11* encodes an ATP binding cassette (ABC) transporter involved in the secretion of a cuticular lipid, which is required for cuticle formation [47,48]. The cuticle layer formed through *ABCG11* mediation acts as a protective sheath against high-stress conditions for vegetative tissues, reproductive organs, embryo epidermis, and the endosperm tissue of developing seeds [47,49,50]. Therefore, the cuticle layer might prevent tissues from losing water during plantlet formation and in the case of *K. pinnata*, slow down leaf drying while plantlets are formed. During germination, seeds undergo a process known as testa rupture after hydration, followed by endosperm rupture and radicle protrusion [51]. During these events, the structures of cellular membranes are damaged, and thus trigger various repair mechanisms [52], possibly those involved in wounding and stress responses. Apart from this, most of these genes are involved in jasmonic acid (JA) synthesis or dependent on JA signaling. Based on the existing literature, JA signaling induces germination during wounding and stress responses [53,54,55]. Therefore, in the case of plantlet formation in *K. daigremontiana* and *K. pinnata*, the formation of indentation and pedestal prior to the emergence of plantlet might be presented as damage to the surrounding tissues, which can in turn, trigger JA signaling to break dormancy and induce plantlet formation.

Genes that are overrepresented in ‘reproduction GO:0000003’ belonged to *K. daigremontiana* clusters Kd4 and Kd8 or *K. pinnata* cluster Kp8. Genes in all three clusters have the same expression pattern, in which the genes were initially downregulated and then upregulated for the subsequent stages (Appendix A, Figure 3). The expression trend indicates that these genes might not be required for the initiation of plantlet formation, but they may be important for plantlet development. It was not surprising to observe the activity of these reproduction-associated genes during early plantlet development, as *K. daigremontiana* plantlet development morphologically resembles embryo development and *LEC1* and *FUS3* were expressed in *K. daigremontiana* plantlets [7]. It was suggested that *K. pinnata* plantlet development only recruits organogenesis, as the expression of *LEC1* was not present in *K. pinnata* leaves [7]. However, this is insufficient to rule out whether other components of the embryogenesis pathway are recruited during *K. pinnata* plantlet formation. Our results suggested that this might be the case, as genes involved in embryo development, such as *ENDOSPERM DEFECTIVE 1* (*EDE1*) and *UNFERTILISED EMBRYO SAC 15* (*UNE15*) were differentially expressed during *K. pinnata* plantlet initiation. Although the specific function of *UNE15* is yet to be demonstrated, it is known to accumulate during late embryogenesis and is associated with stress response [56]. As for *EDE1*, it was shown to interact with microtubules to regulate the formation of *Arabidopsis* endosperm and embryo [57]. The upregulation of these genes in subsequent stages of plantlet development was particularly intriguing, since, in the later stages of harvested tissues, the cotyledons were already present (Figure 1), indicating the seedling-equivalent-stage of plantlet maturity. The only plausible explanation is that these genes might have different functions during the seedling stages. Further research into the role of these genes during plantlet formation and whether other embryogenesis genes are involved in the process is required to obtain conclusive evidence on the participation of embryogenesis during *Kalanchoë* plantlet formation.

The GO term ‘regulation of hormone levels GO:0010817′ includes genes that are involved in the regulation of hormone levels. These genes belong to *K. daigremontiana* clusters Kd5 and Kd6 and *K. pinnata* cluster Kp12. The gene clusters in Kd5 and Kp12 exhibit the same expression pattern as the genes in the reproduction GO term, which is downregulated between the first two stages, then continously increased in expression in subsequent stages. However, the gene cluster in Kd6 continued to rise in expression since the beginning (Appendix A, Figure 3). These genes might be more involved in the latter stages of plantlet formation in *K. daigremontiana*. Genes controlling hormone levels are likely to be involved in plantlet formation, as hormones are known to play a major role in plant growth and development [58]. The plant hormone auxin has been extensively studied and is known to affect various aspects of plant development [59]. Our data show that most of the genes in these GO terms are involved in auxin transport (*PIN-FORMED 1* [*PIN1*], *AMINOPEPTIDASE M1* [*APM1*], *PINOID* [*PID*], *TORNADO 1* [*TRN1*], *PATELLIN PROTEIN 5* [*PATL5*], *WRKY DNA-BINDING PROTEIN* [*WRKY23*], auxin biosynthesis (*STYLISH 1* [*STY1*]) or are regulated by auxin (*SHI-RELATED SEQUENCE 5* [*SRS5*], *SRS7*) [60,61].

The last GO term selected for further analysis is ‘response in karrikin GO:0080167′, in which only three genes, such as *GLYCEROL-3-PHOSPHATE SN-2-ACYLTRANSFERASE* (*GPAT1*), *GIR2*, *MYB DOMAIN PROTEIN 94* (*MYB94*) overlapped between the two gene clusters, showing an overrepresentation of this GO term. Karrikins are known to trigger seed germination and seedling establishment [62,63]. Therefore, it is also possible that the expression of genes in these clusters indicate that initial seed germination and plantlet development share genetic pathways. Existing studies showed that *GPAT1*, *GIR2*, and *MYB94* function differently (Appendix A). However, both genes were overrepresented in both species in a similar trend, suggesting that these genes might have similar or complementary functions during plantlet development. *GIR2* promotes histone deacetylation to regulate root hair development. *MYB94* inhibits auxin-induced callus formation mediated via a root developmental pathway and *GPAT1* is involved in the differentiation of tapetal cells, which are cells in anthers [64,65].

Gene ontology analysis revealed that additional unique GO terms are associated with each of the species studied, possibly (Appendix A) since *K. daigremontiana* and *K. pinnata* have different modes of plantlet formation [7,32]. In the case of *K. daigremontiana*, five GO terms were overrepresented in gene clusters 2 and 7, exhibiting a similar expression pattern to trend 1. This indiactes that these genes were upregulated in the S1 stage of plantlet formation when compared with young leaf margins. Then, the downregulation of these genes continued across the subsequent plantlet developmental stages (Figure 2). Apart from ‘response to radiation GO:0009314’, overrepresentation of these GO terms was expected, as these terms signify general processes that occur during plantlet development, since samples contained developing young leaves and plantlets. The GO term ‘response to radiation GO:0009314’ might have indicated that as developing leaves mature, the plants actively respond to electromagnetic radiation, including light stimulus. The developing leaves might be responding to radiation to obtain sufficient light for growth, but also for protection from radiation damage [66,67]. At the same time, the plants might be detecting whether there is sufficient light exposure for plantlet formation as plantlet formation occurs only under long-day conditions [14,26].

The GO terms that share a similar expression pattern to trend 3 belong to gene clusters 4, 5, and 8. Genes in these GO terms have the exact opposite expression trend as previously mentioned for other GO terms. These genes exhibit downregulation from the young leaf margin stage or S1 plantlet stage, and then upregulation across subsequent plantlet formation stages (Figure 3). Interestingly, The GO term ‘cellularization GO:0007349’ was found to be uniquely overrepresented in cluster 4 in *K. daigremontiana*. In plants, cellularization is the process in which the multi-nucleated syncytium separates into individual cells and develops into seed endosperm [68]. In plantlet formation, cellularization occurs only during the later stages of plantlet formation. The presence of ‘carbohydrate transport GO:0008643’ in young developing leaves and plantlets may be linked with carbohydrate requirements during plantlet formation, which is also known to regulate plant–pathogen interaction [69,70]. This GO term was found to be upregulated from the development of young leaf margins until the final stage of plantlet formation.

The downregulation of ‘plastid organization GO:0009657’ in *K. daigremontiana* suggests that the arrangement of plastids is not necessary for developing young leaves and plantlets [71].

*K. pinnata* clusters 1 and 7 include genes that were upregulated at the leaf notches 4 h after leaf detachment, but were gradually downregulated after this stage. GO terms that were overrepresented in these clusters included ‘immune system process GO:0002376′, ‘response to drug GO:0042493’, and ‘response to oxygen levels GO:0070482’. In accordance with the GO database AmiGO2, the GO term ‘immune system process GO:0002376’ refers to the immune system in response to potential internal or invasive threats caused by both biotic and abiotic factors. This term was overrepresented possibly due to the wounding caused by leaf detachment. The same explanation might apply to the overrepresentation of ‘response to oxygen levels GO:0070482’. Changes in oxygen level might have occurred during the process and triggered *K. pinnata* plantlet formation. A previous study has shown that oxidative stress imposed by nitric oxide can affect *K. pinnata* plantlet formation [35]. The plantlets of *K. tubiflora,* a constitutive-plantlet-forming species, have lower antioxidant defence compared with the mother leaves [36]. Unique overrepresented GO terms for *K. pinnata* in clusters 3 and 6 included upregulated genes at the initial stages, downregulated after 4 h of leaf detachment, and upregulated after 24 h. The GO term ‘defence response GO:0006952’ usually denotes a response to an injury, which results in structural damage to the organism that might be leaf detachment in this case. The GO term ‘regulation of biosynthetic process GO:0009889’ includes genes that mediate the synthesis of substances, probably as a result of the metabolism of carbohydrates to retrieve energy for plantlet initiation. The overrepresentation of ‘drug metabolic process GO:0017144’ was unusual. However, bufadienolide compounds that have anticancer and antiviral effects are present in the leaves of *K. pinnata.* This term might be overrepresented as a result of the degradation of these compounds following the leaf detachment.

Gene clusters 8 and 12, which exhibited a similar expression pattern to trend 3 in *K. pinnata,* include downregulated genes that were upregulated after leaf detachment and remained upregulated in subsequent time points. Overrepresented GO terms in these clusters were ‘response to inorganic substance GO:0010035’ genes, which might have changed the expression in response to water deprivation after the leaf was excised from the mother plant. The GO term ‘wax biosynthetic process GO:0010025’ contained genes that possibly play a role in preventing water evaporation from the removed leaves, as it can be seen that plantlets appear after 9 days of leaf detachment. The GO term ‘stem cell population maintenance GO:0019827’ contained genes that were also found to be overrepresented in trend 3. Stem cells in plants are usually maintained in the SAM, RAM, and vascular meristems for growth, as plants develop post-embryonically [72]. Moreover, these cells contribute to the regeneration of lost organs through organogenesis routes due to biotic or abiotic stress [73]. At a cellular level, growth, development, and regeneration share the same genetic pathways [74]. The epiphyllous buds in *K. pinnata* require the presence and maintenance of a stem cell niche from which plantlets will emerge. The GO term ‘cell wall organization or biogenesis GO:0071554’ found in cluster 12 could feasibly play an important role during cell division, leading to the generation of new plantlets. The GO term ‘response to auxin’ was also overrepresented and included genes involved in the organogenesis pathway of plantlet formation. Auxin plays a key role in essential developmental processes in plants, such as embryogenesis, gametogenesis, vascular patterning, and flowering [59]. Auxin accumulation promotes lateral organ initiation in the SAM, and is carried via polar transport, facilitated by the PIN1 protein [75]. Auxin signaling in stem cells is mediated by *AUXIN RESPONSE FACTORS* (*ARFs*) to positively regulate *CLAVATA 3* (*CLV3*) in the CZ of the SAM [76]. It has been demonstrated that auxin plays a key role in plantlet formation in *K. marnierianum* [24]. Upregulation of ‘plant organ formation GO:1905393′ genes was found after leaf excision, possibly facilitating plantlet initiation. The expression of organogenesis genes in subsequent time points after leaf detachment was expected, as inducible-plantlet-forming species are known to form plantlets through organogenesis routes [7].

The GO term ‘positive regulation of seed germination GO:0010030’ includes genes in trend 6, which were upregulated after leaf detachment and downregulated 48 h after leaf detachment. This term is involved in the activation of seed germination processes. This suggests that *K. pinnata* plantlet formation activates the same pathways recruited in seed germination. Interestingly, the presence of seed and embryo genes has only been reported for *K. daigremontiana* constitutive-plantlet-forming species [7]. The GO terms ‘cellular response to endogenous stimulus GO:0071495’ and ‘shoot system development GO:0048367’ were uniquely overrepresented in *K. pinnata*. These genes follow trend 8, where upregulation occurs only 48 h after leaf detachment. Signals to the epiphyllous buds from within the plant and meristematic activity in the buds are expressed simultaneously on *K. pinnata* inducible-plantlet-forming species. When epiphyllous buds initiate plantlet formation, the first visible structure is the SAM, and it becomes visible 9 days after the leaf was excised from the mother plant. Surprisingly, the GO term ‘shoot system development GO:0048367’ is present only 2 days after inducing plantlet formation in *K. pinnata*. Plantlet formation in *K. pinnata* is activated by the detachment of leaves. In accordance with our data, within the first 4 h, the set of upregulated genes which is possibly more relevant to the vegetative reproductive process was ‘immune system process GO:0002376’, ‘defence response GO:0006952’, ’regulation of biosynthetic process GO:0009889’, ‘wax biosynthetic process GO:0010025’, ‘cellular component organization or biogenesis GO:0071840’. These GO terms indicated the sensing of mechanical damage to the plant integrity after leaf excision.

After the plant recognized and responded to leaf damage, the set of genes that upregulated within 24 h after leaf detachment included ‘stem cell population maintenance GO:0019827’, ‘cell wall organization or biogenesis GO:0071554’, ‘multicellular organismal reproductive process GO:0048609’, and ‘plant organ formation GO:1905393’. The upregulation of these genes 24 h following leaf detachment from the mother plant could indicate that the epiphyllous buds were at this point already initiating an organogenesis program to form plantlets. Furthermore, the GO term ‘shoot system development GO:0048367’, which includes specific organogenesis genes designated to shape the SAM was upregulated 48 h after leaf detachment.

## 4. Materials and Methods

### 4.1. Plant Materials and Growth Conditions

Wild-type *K. daigremontiana* plants were grown in SANYO versatile environmental test chamber MLR-351 at 23 °C with a photoperiod of 16/8 h with 50 μmol m^−2^s^−1^ light and 60% humidity. Wild-type *K. pinnata* plants were grown in Percival Scientific growth chamber AR-60L at 23 °C with a photoperiod of 8/16 h with 30 μmol m^−2^s^−1^ light and 60% humidity. The plants were grown in a mixture of six parts with Levington^®^ F2 Seed and Modular Compost (The Scotts Company, Rustington, UK), one part Vermiculite V3 medium (Sinclair Pro, UK), and one part Perlite P35 standard (Sinclair Pro, Ellesmere Port, UK). Four distinct stages of plantlet formation in wild-type *K. daigremontiana* were identified to include stages of plantlet initiation (Figure 1A). A leaf exhibiting at least three of these stages of plantlet maturation along its leaf margin was carefully selected for use. The leaves were removed using a razor blade and 0.3 cm^2^ tissues at the leaf notches were harvested using the blade. The control samples consisted of the whole margins of 1–2 cm long leaves when measured from base to tip of each leaf. For *K. pinnata*, four time points after leaf detachment (0, 4, 24, 48 h) were selected. Moreover, 0.3 cm^2^ tissues at the leaf notches were harvested with a razor blade from the unattached leaves (0 h) and leaves detached after 4, 24, and 48 h. The mid-section of the leaf (48 h after leaf detachment) was also used as an additional control. No major morphological changes were present during these time points. All of the harvested samples were immediately frozen in liquid nitrogen and stored at −80 °C until RNA extraction. Images in Figure 1A were taken using an S8AP0 Stereo Microscope (Leica Microsystems, Wetzlar, Germany) with a Digital D3100 camera (Nikon, Tokyo, Japan) attached. Images in Figure 1B were taken using the same microscope, but with a GX-CAM-Eclipse camera (GT Vision, Wickhambrook, UK) attached. All of the images were processed with ImageJ 1.48v to include a scale bar.

### 4.2. RNA Extraction and RNA-Sequencing

Total RNA from each sample was extracted with a Qiagen RNeasy Plant Mini Kit (Qiagen, Manchester, UK), in accordance with the manufacturer’s protocol with modification. For *K. daigremontiana* samples, 600 μL of RLC buffer with 10 mg of polyvinylpyrrolidone (PVP) with a molecular weight of 40,000 was used for a maximum of 100 mg tissue powder. For *K. pinnata* samples, 600 μL of RLC buffer with 10 mg of polyethylene glycol (PEG) with a molecular weight of 40,000 was used for a maximum of 100 mg of tissue powder. The RLC buffer was supplemented with PVP or PEG to obtain the highest total RNA yield and purity for each of the species. Then, the mixed solution of samples from *K. daigremontiana* or *K. pinnata* was vortexed and incubated for 1 min at 56 and 80 °C, respectively prior to following the subsequent steps of the kit’s protocol. Purified RNA samples were sent for Illumina HiSeq 2000 sequencing technology (The University of Manchester Sequencing Facility). Quality checks on the RNA-sequencing reads were performed with FastQC (Babraham Bioinformatics, Cambridge, UK). Annotated RNA-sequencing data are available from ArrayExpress (ID: E-MTAB-11794). Reads were quality trimmed using Trimmomatic_0.36 (PMID: 24695404). Since the genomes were not available for *K. daigremontiana* and *K. pinnata,* RNA-sequencing reads were mapped with the *K. fedtschenkoi* genome v1.1 (https://phytozome.jgi.doe.gov/pz/portal.html; last accessed on 10 June 2022) using STAR_2.5.3a (PMID: 23104886). Counts per gene were calculated with HTSeq v0.6.1 (PMID:25260700) using annotation from *K. fedtschenkoi* genome v1.1 (https://phytozome.jgi.doe.gov/pz/portal.html; last accessed on 10 June 2022). The normalization, principal components analysis, and differential expression were calculated with DESeq2_1.16.1 (PMID:25516281).

### 4.3. Expression Profiling and Clustering Analysis

Based on the differential gene expression readings, volcano plots (Figure 5) were subsequently generated using R_3.4.1 to show the proportion of genes that are significantly upregulated or downregulated between samples of different stages or time points. The dataset was filtered using adjusted *p*-value ≤ 0.05, and log2 fold-change > |0.6| for *K. daigremontiana* and log2 fold-change > |1.585| for *K. pinnata*. The means were calculated for each condition on a log scale and Z-transformed (a normalization where for each gene the average of the 5 means was set to zero and the standard deviation was set to 1). Then, a heatmap was generated with DESeq2 to cluster genes with similar expression profiles (Figure 2). The number of clusters was determined based on different expression trends between time points and stages. The filtered dataset was used to identify genes that are differentially expressed from one stage to another. Venn diagrams (Figure 4) showing the number of differentially expressed genes and whether these genes overlapped between different comparisons were generated using http://bioinformatics.psb.ugent.be/webtools/Venn/ (last accessed on 10 June 2022). The average expression level of genes from each cluster was used to generate line graphs using Microsoft Excel to illustrate changes in the expression pattern of genes in each cluster. Gene clusters with similar trends were collated on the same graph and presented in Figure 3.

### 4.4. Gene Ontology Enrichment Analysis

Gene ontology (GO) enrichment analysis was performed on each of the gene clusters from *K. daigremontiana* and *K. pinnata* using the best *Arabidopsis thaliana* homologue (https://phytozome.jgi.doe.gov/pz/portal.html; last accessed on 10 June 2022). The list of genes in each cluster was analyzed for biological processes that are overrepresented using http://geneontology.org/ (last accessed on 10 June 2022). ReviGO was used to remove redundant GO terms that are enriched in each cluster. A multiple list comparator tool (http://www.molbiotools.com/listcompare.html; last accessed on 10 June 2022) was used to determine whether overrepresented terms with 0% dispensability from each cluster overlap or are exclusive to each species. The overlapping GO terms are presented in Appendix A. GO terms with more specific biological functions that are more relevant to plantlet formation were selected. Genes present in selected GO terms are presented in Appendix A. The number of genes in these GO terms and whether these genes overlap between different gene clusters were generated using http://www.molbiotools.com/listcompare.html (last accessed on 10 June 2022) and are shown in Figure 2. GO terms exclusively present only in one species and not the other are shown in Appendix A.

## 5. Conclusions

This study is the first global transcriptome analysis of plantlet formation. Based on the different modes of plantlet formation and plantlet morphological structures, we successfully selected tissues with almost exclusively distinctive plantlet stages and time points to conduct our experiment. Clustering of biological replicates of our tissue samples signifies that our results are very consistent. Our data suggest that plantlet formation in *K. daigremontiana* and *K. pinnata* are largely unique, as suggested by the greater number of unique genes and GO terms overrepresented in each species. However, overrepresentation of the same GO terms in both species suggests that plantlet formation in *K. daigremontiana* and *K. pinnata* relies on the participation of pathways involved in signaling, wounding response, hormone regulation, reproduction, and response to karrikin. Yet, it is not clear how the unique GO terms recruited during plantlet formation are unique to each species. Our findings remain preliminary and still require extensive validation and experiments to understand the molecular mechanisms involved in plantlet formation. Nonetheless, our analysis will be a pioneer for future research on *Kalanchoë* plantlet formation. Greater knowledge in this field will accelerate our understanding of the various asexual reproduction strategies in plants.

## Figures and Tables

**Figure 1 plants-11-01643-f001:**
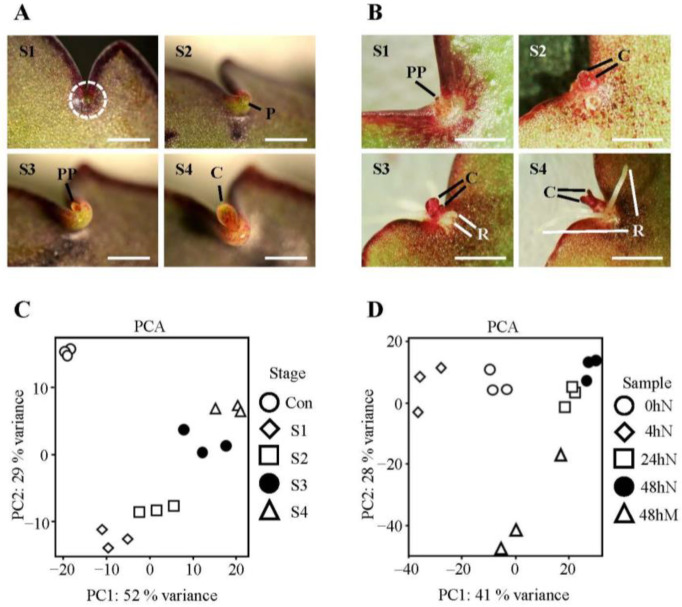
*K. daigremontiana* and *K. pinnata* plantlet analyses and principal component analysis (PCA). (**A**) Four distinctive stages of plantlet formation in *K. daigremontiana* were selected for the RNA-sequencing experiment. (Stage 1, S1) leaf notch without pedestal formation; (Stage 2, S2) leaf notch with pedestal formation; (Stage 3, S3) leaf notch with pedestal and with an emerging plantlet primordium; (Stage 4, S4) leaf notch with plantlet primordium with visible cotyledons. (**B**) Plantlet formation in *K. pinnata*. (S1) plantlet primordium emerging from leaf notch; (S2) plantlet primordium with visible cotyledons emerging from leaf notch; (S3) plantlet with emerging root primordia; (S4) plantlet with extended root formation. The scale bar is 1 mm. C: Cotyledon; P: Pedestal; PP: Plantlet primordium; R: Root. (**C**,**D**) PCA of RNA samples harvested from leaf notches of *K. daigremontiana* at stages (**A**) and of *K. pinnata* at time points 0 (h), 4, 24, and 48 h. Margins of young 1–2 cm leaves were used as control samples (Con) for *K. daigremontiana*, whereas the control samples for *K. pinnata* were the mid-section of detached leaves after 48 h. Three biological replicates were generated for each developmental stage or time point. Con: Control; N: Leaf notches; M: Leaf mid-section.

**Figure 2 plants-11-01643-f002:**
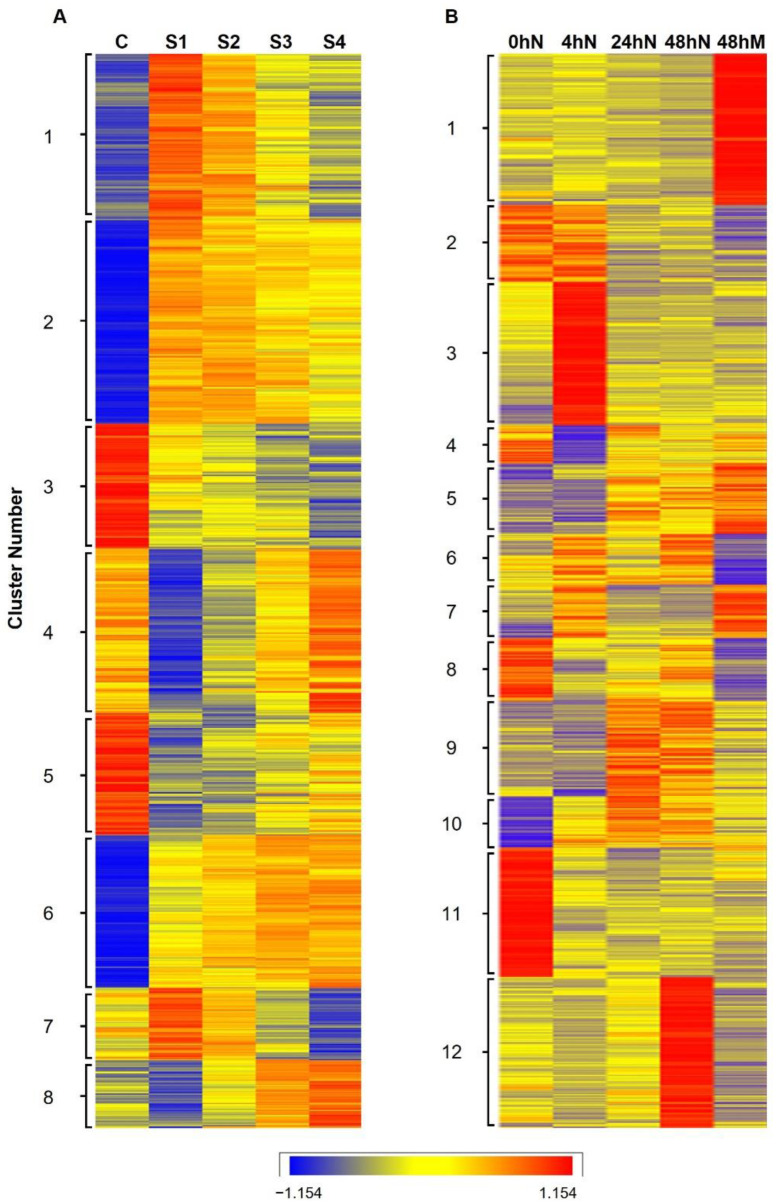
Heatmap shows hierarchical clustering of genes with similar expression profiles. (**A**) The heatmap shows a total of 4594 differentially expressed genes in *K. daigremontiana*, which are grouped into eight clusters with similar expression profiles. Only genes with adjusted *p*-value ≤ 0.05 and log2 fold-change > |0.6| were selected. C: Control; S1: Stage 1; S2: Stage 2; S3: Stage 3; S4: Stage 4. (**B**) The heatmap shows a total of 5706 differentially genes in *K. pinnata*, which are grouped into 12 clusters with similar expression profiles. Only genes with adjusted *p*-value ≤ 0.05 and log2 fold-change > |1.585| were selected. N: Leaf notches; M: Leaf mid-section. The color range of blue-ye-low-red indicates expression levels from low to high.

**Figure 3 plants-11-01643-f003:**
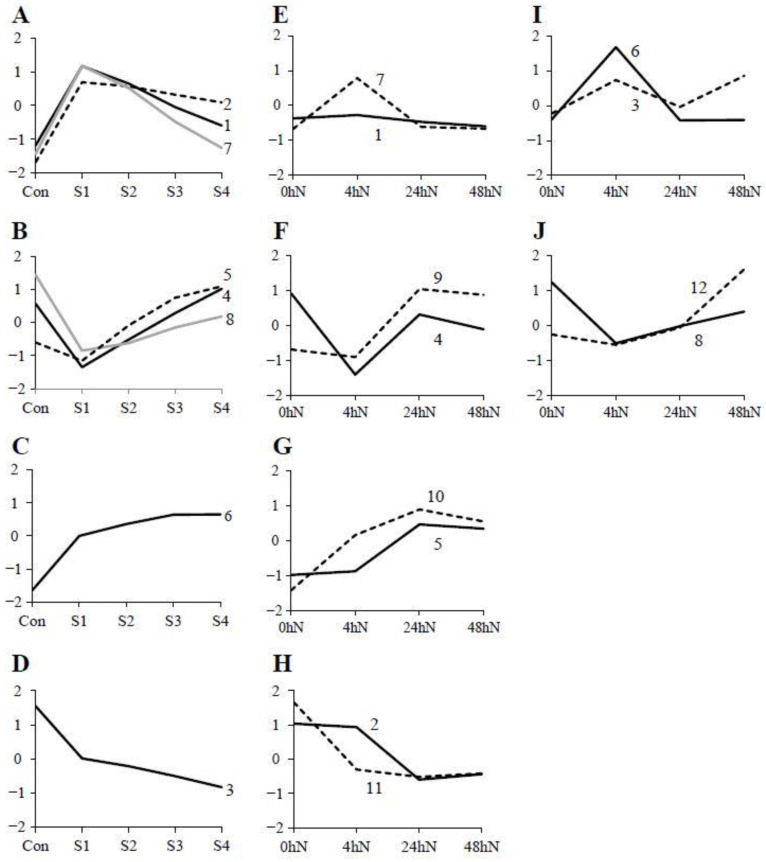
Graphical representation of average change in expression of genes in different gene clusters. (**A**–**D**) The expression pattern of eight gene clusters in *K. daigremontiana* across different plantlet developmental stages. (**E**–**J**) The expression pattern of twleve gene clusters across different time points after detachment of *K. pinnata* leaf. Clusters of genes with a similar trend of changes in expression are visualized on the same graph. The number adjacent to each graph line corresponds to the number of the gene cluster. Con: Control; S1: Stage 1; S2: Stage 2; S3: Stage 3; S4: Stage 4; N: Leaf notches.

**Figure 4 plants-11-01643-f004:**
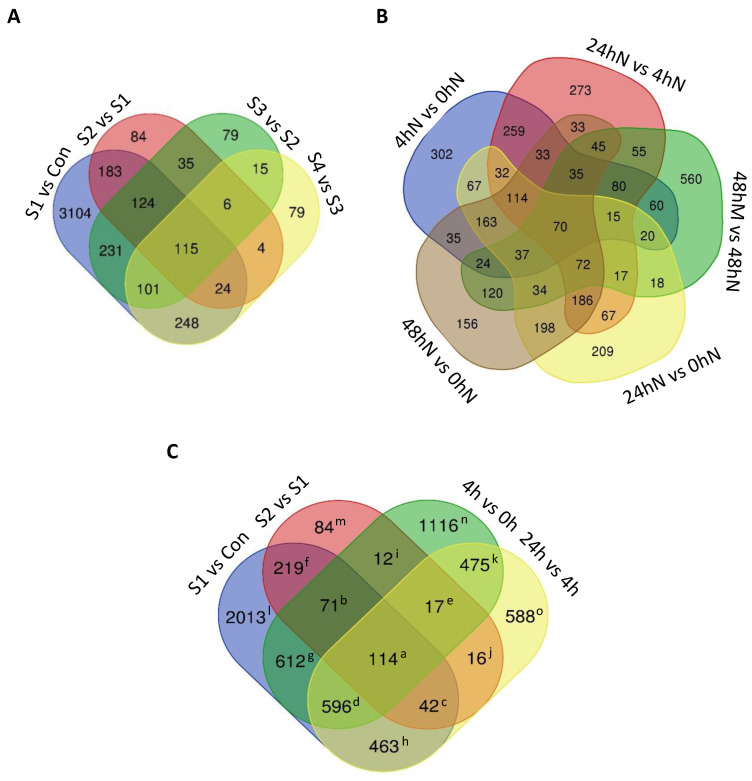
Venn diagrams showing the number of differentially expressed genes (DEGs) during plantlet formation. The number of DEGs overlapped from a comparison of (**A**) different plantlet developmental stages from *K. daigremontiana* or (**B**) samples harvested at specific time points upon *K. pinnata* leaf detachment. (**C**) The number of exclusive and overlapping differentially expressed genes between selected plantlet stages from (**A**) and time points from (**B**). Superscript alphabets correspond to the list of genes in Appendix A Con: Control; S1: Stage 1; S2: Stage 2; S3: Stage 3; S4: Stage 4; N: Leaf notches; M: Leaf mid-section.

**Figure 5 plants-11-01643-f005:**
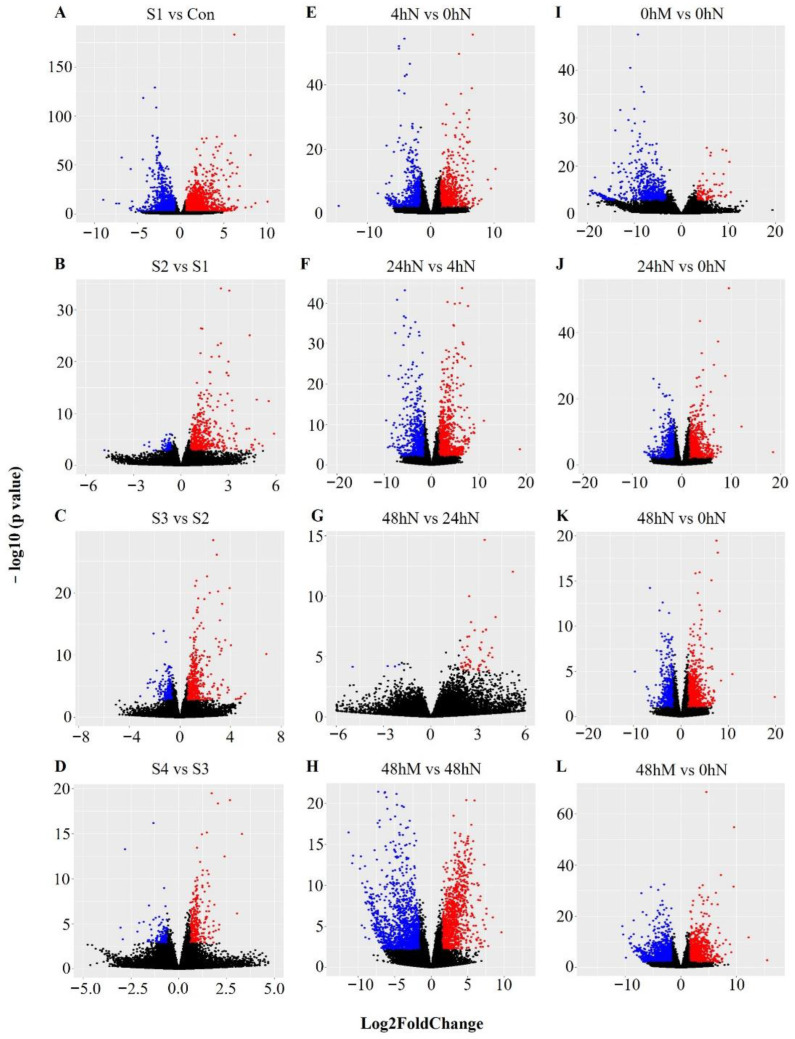
Volcano plots showing statistical significance of changes in gene expression and their expression fold-change during plantlet formation. Each plot shows a comparison between two developmental stages of plantlet formation in *K. daigremontiana* (**A**–**D**) or between two time points post-detachment of *K. pinnata* leaves (**E**–**L**). Each plot shows −log10 (*p*-value) against log2 fold-change. Blue: False discovery rate (FDR) ≤ 0.05, log2 fold-change ≤ −0.6 for (**A**–**D**), log2 fold-change ≤ −1.585 for (**E**–**L**); Red: FDR ≤ 0.05, log2 fold-change > 0.6 for A–D, log2 fold-change > 1.585 for (**E**–**L**); Black: Not significant. Con: Control; S1: Stage 1; S2: Stage 2; S3: Stage 3; S4: Stage 4; N: Leaf notches; M: Leaf mid-section.

**Figure 6 plants-11-01643-f006:**
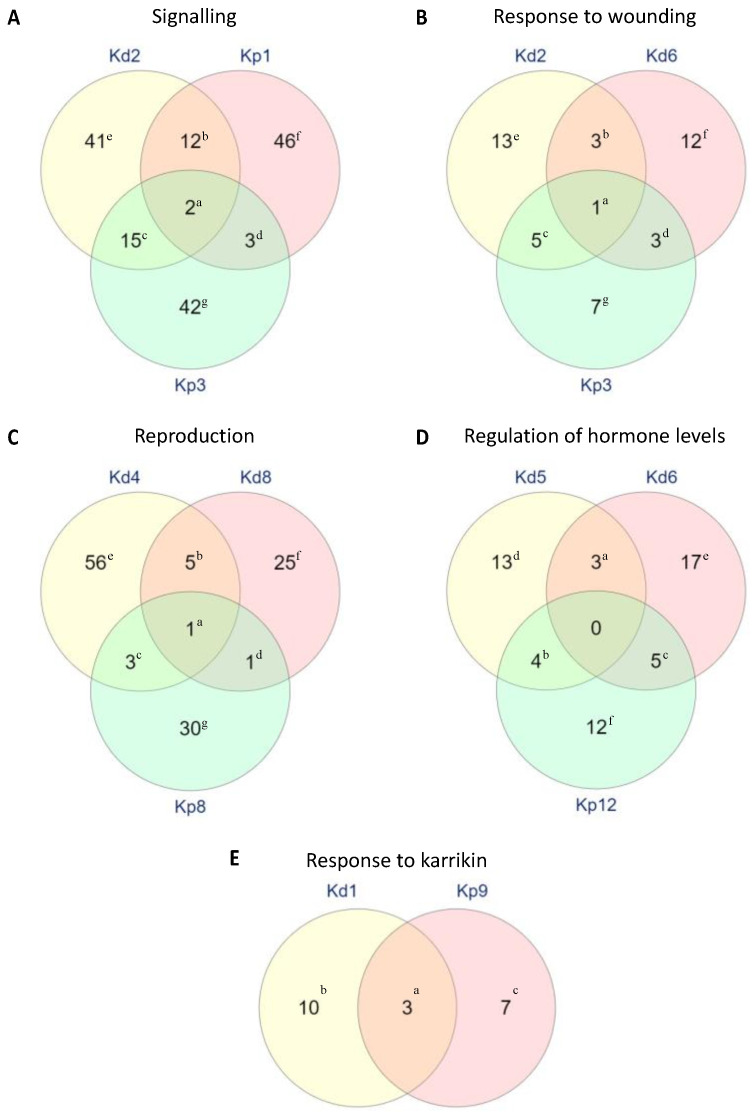
Number of genes in selected GO terms that are overlapped between different clusters of two Kalanchoë species, *K. daigremontiana* (Kd) and *K. pinnata* (Kp). Genes in signalling (**A**), response to wounding (**B**), reproduction (**C**), regulation of hormone levels (**D**) and response to Karrikin (**E**). The numerical value next to the species name symbol (Kd, Kp) represents the corresponding gene cluster in Figure 2. The numerical value in the Venn diagrams represents the number of genes. Superscript alphabets correspond to the list of genes in Appendix A.

## Data Availability

Not applicable.

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
