# Peer review of "Comparative Transcriptome Analysis of Two Kalanchoë Species during Plantlet Formation"

_plants, 2022, doi:10.3390/plants11131643_

Round 1

Reviewer 1 Report

The manuscript describes the comparative transcriptome of Kalanchoë species with different types of plantlet formation (i.e. continuous and inducible) and is mostly well written.

However, there are parts/paragraphs which did not find their place. F.e.:

L118-L134: this part is rather a summary and can be the part of the discussion/conclusion, but surely not the introduction.

L159-L167: This should be the part of the materials and methods, as this is not result.

L315-324: This is – at least partly –the description of the Suppl.Table 1, so it should be removed from the text of the results. But what should be described here what can be seen from the table.

Others:

L140: What kind of “specialized structure”? Does it has a name?

Anyway, what is the morphological-physiological difference between this “specialized structure” of K. daigremontiana and “bud-like structure” of K. pinnate? Whether there is any difference?

Molecular, morphological and physiological traits should be evaluate together, otherwise, I am not sure what is the exact proof that not somatic embryogenesis occurs in both cases (even if in a bit different way)

Please, use the terms and expressions with the same way throughout the manuscript. For example: signaling or signalling? etc. Check the text!

Reviewer 2 Report

I have read, carefully and with great interest, the manuscript entitled „Comparative transcriptome analysis of two Kalanchoë species during plantlet formation” submitted for publication in Plants MDPI. The aim of this research was to study, for the first time, biological processes and genes involved during plantlet formation in two Kalanchoë species differing in the plantlet formation strategy.

In my opinion the manuscript will be interested for Plants readers. The obtained results are important for undrestanding molecular mechanism of plantlet formation, however major corrections/changes are recommended to improve the quality of the manuscript before final publication.

Generally, the manuscript is too long and should be shortened, especially the Introduction and Discussion parts. The number of the cited references should be significantly reduced. There are some the same information repeated in different manuscript parts – it should be eliminated.

Abstract

The full Latin name of the species with authors initials should be provided (the species is described for the first time in abstract, so it is neccesary).

Introduction – this chapter is too long, moreover the last paragraph should describe more precisely the study aims, but without repeating the description concernigs results and conclusions.

Materials and methods

The Authors should explain, why different conditions of plant cultivation were chosen? The differences in light conditions, in my opinion, could influence the gene expression in tested species, and finally moderate the obtaind results.

Line 693 – it is Fig. 5.1A, it should be Fig. 1A.

Lines 699-700 – information concerning the control for K. pinnata is missing. It is also missing in figures.

Lines 709-713 – Why did the Authors use different protocols for RNA extraction for tested species?

Round 2

Reviewer 2 Report

I accept the manuscript after revision.